# Going Deeper with General and Specific Inductive Bias for Real-Time Stereo Matching

## Abstract

Inductive Bias (IB) has sparked a revolutionary transformation by incorporating the advantages of CNNs and Transformers, including scale invariance and integration of locality and long-range dependencies, which is called general IB for its wide applicability. However, its efficacy is currently not enjoyed by stereo matching, one of the geometric vision tasks, because of the ignorance of volume-level scale invariance and the limitation of high real-time requirement. In contrast, a specific IB is adopted by constructing volume structure in stereo matching task, which helps to finally generate a confidence volume to predict disparity map (output), but fewer studies go into the specific volume structure. Based on the above issues, this paper develops a novel model named UStereo to introduce the general IB to stereo matching. Technically, we adopt inter-layer fusion to break down volume-level scale invariance to a recurrence strategy in initialization for information at low resolution and refinement process for the high, which further extends to capture long-range dependencies after shallow stacks of convolutions and normalization without time-consuming Transformers. Additionally, to reveal the role that the volume structure constructed by specific IB plays during inference, we propose the first-time in-depth study of volume at low resolution through varying degrees of restraint as well as 3 original statistic indicators to reflect the characteristics of representation within volumes. Experiments demonstrate UStereo has competitive performance with both fast speed and robust generalization, and ablation studies show the effectiveness of introducing general IB. Moreover, our analysis of the volumes at low resolution suggests they can be viewed as confidence volumes and a concentrated distribution of the disparity within volumes leads to enhanced performance, which could extend the role of the specific IB.

## 1 Introduction

Generally speaking, different inductive biases (IB) or preference can enhance the capacity of a given network to model different forms of information and improve its generalization ability (Wolpert et al., 1995; Baxter, 2000; Raghu et al., 2021; Bommasani et al., 2021; Goyal & Bengio, 2022), which is seen as a viable way to achieve both efficiency and effectiveness (Zhang et al., 2022; Wan et al., 2023). The most discussed IB today revolves around CNNs and Transformers *i.e.* scale invarience IB from CNNs and long-range dependencies from Transformers that compensate for the locality in CNNs, which we call **general IB** in this paper for its universality for almost vision tasks (Liu et al., 2021; Xu et al., 2021b; Ren et al., 2022). However, so far efficacy of the general IB has not been enjoyed by certain low-level geometric vision tasks such as stereo matching.

Stereo matching task estimates a dense disparity or depth map from a pair of stereo images under the epipolar geometry constraint (Scharstein & Szeliski, 2002). It is a classic and important vision problem that has been studied for almost half a century (Marr & Poggio, 1976), and is wildly used in robotics (Schmid et al., 2013), SLAM (Gomez-Ojeda et al., 2019), autonomous driving(Menze & Geiger, 2015), computer assisted surgery (Allan et al., 2021), *etc.* Notably, efficiency and real-time performance are crucial requirements in these applications.

For the balance between the effectiveness and efficiency in stereo matching, mainstream works adopt CNN-based operation and volume structure. Despite the use of CNNs, current model structures for stereo matching neglect the volume-level scale invariance since they usually use a U-shape

network at certain volume or adopt Coarse-to-Fine (CTF) strategy to generate disparity by refining the disparity map during inference (details see §2). For long-range dependencies to compensate for the locality in CNNs, the task-specific geometry constraint and real-time requirement limit the use of Transformers to capture long-range dependencies for the computing burden in cross- and self-attention mechanisms (Li et al., 2021). Therefore, how to capture long-range dependencies easily and effectively without Transformers becomes an important problem. As for the volume structure, it is inspired by the task-specific geometric constraint in stereo matching and used to generate a confidence volume at the final output to predict the disparity map, which could be viewed as a **specific IB** for stereo matching. Though volume-based methods have become the de-facto method, fewer studies go deeper into the power of the specific IB *i.e.* the volume structure (Eq.1).

Analysis of current approaches naturally leads us to the questions: (i) How we can adopt the general IB (scale-invariance and long-range dependencies) to stereo matching and keep real-time. (ii) What roles of specific IB can play such as enhancing performance without altering the model architecture, or determining the underlying characters of representation that a high-performing model can learn.

Our contributions are as below:

- General IB and UStereo: We design a new model called UStereo. Disparity generation during inference is given up and replaced by direct inter-layer fusion. Therefore, volume-level scale invariance is broken down into two stages, initialization and refinement process. A novel proposed fusion strategy **D**ense **S**cale-Aware **F**usion (DSF) establishes a dense connection with volumes at high resolution in recurrence form, and information at low resolution is integrated during refinement process, which extends to a new block **M**ixed **D**irect **L**ong-Range **C**ompensation (MDLC) to capture long-range dependencies by its inherent volume structure after shallow stacks of convolutions and normalization without time-consuming Transformers.
- Specific IB and Representation: We conduct an in-depth study of the volume structure at low resolution, deep into the role of specific IB. We propose two deep supervision with strong and relaxing restrictions respectively to concentrate the distribution of the disparity (representation) within the volume. Additionally, a novel deep self-supervision is proposed which controls the distribution of disparity. We also adopt three original statistic indicators related to the concentration degree for the volume and observe the distribution along the disparity dimension of volume at low resolution. To our best knowledgement, it is first-time to study the specific IB and representation of volumes.
- Experiments: UStereo achieves competitive results to other volume-based algorithms with fast speed and shows generalization capabilities across virtual and real datasets, as well as diverse natural scenes and small abdominal cavity scenes. Additionally, our analysis of the volumes at low resolution suggests they can be viewed as confidence volumes and a concentrated distribution of the disparity within volumes leads to enhanced performance, which could extend the role of the specific IB.

## 2 RELATED WORK

**The Specific Inductive Bias for stereo matching** Constrained by geometric relationship, in which only pixels in certain disparity search space are computed in need (in Fig.1), most works in stereo matching have volumes structure by Eq.1, where $\mathcal{B}(\cdot, \cdot)$ can be full correlation (Mayer et al., 2016; Tonioni et al., 2019; Duggal et al., 2019; Lipson et al., 2021; Song et al., 2020; Xu & Zhang, 2020), concatenation (*Cat*) (Kendall et al., 2017; Chang & Chen, 2018; Zhang et al., 2019; Yang et al., 2020b) or group-wise correlation (*Gwc*) (Guo et al., 2019; Xu et al., 2021a; Yao et al., 2021; Xu et al., 2023). The volume structure helps to generate a confidence volume at the final output for a model in stereo matching to predict a disparity map and it could be a specific IB for stereo matching:[1]

$$\boldsymbol{V}_{x,y,d,:} = \text{Concat}_i(\mathcal{B}(\boldsymbol{f}^l_{x,y,(\frac{c}{g}\times i:\frac{c}{g}\times(i+1))}, \boldsymbol{f}^r_{x-d,y,(\frac{c}{g}\times i:\frac{c}{g}\times(i+1))}))$$

$$d \in [0, \min(d_{max}, x)], \ 0 \le i \le (g-1), \ i \in \mathbb{N} \text{ and } g \in \mathbb{N}^+ \tag{1}$$

where $d$, $c$ and $g$ represent, disparity, feature channel number for left/right feature, $\boldsymbol{f}^l/\boldsymbol{f}^r$, and group number for $Gwc$ respectively. The symbol ":" represents the indexing mechanism similar to that in computer languages. $\text{Concat}_i$ means concatenating processed features by $\mathcal{B}$.

---

[1] We use $\boldsymbol{D}$, $\boldsymbol{V}$, $\boldsymbol{V}_{Gwc}$, $\boldsymbol{V}_{Cat}$ denote any disparity map, volume and $Gwc$ and $Cat$ volume. For the extra dimension in 4D volume compared to 3D volume, we call it $\beta$ dimension (3D could be viewed as $\beta = 1$).

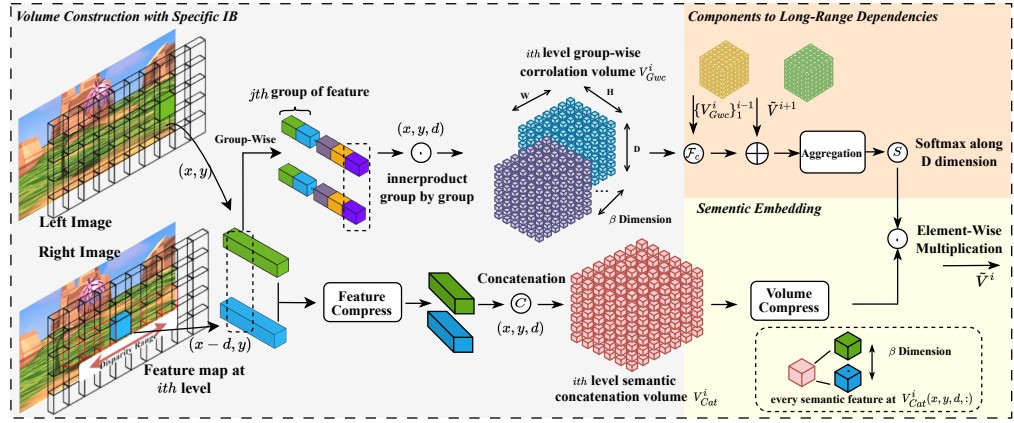

Figure 1: The schematic illustration of the volume construction and our proposed block Mixed Direct Long-Range Compensation which is composed of components to long-range dependencies and semantic embedding.

**Model Strategy for Stereo Matching**  There are two mainstream model strategies to generate confidence volume, namely U-Shape Net (in Fig.2 left *i.e.* Hourglass) (Chang & Chen, 2018; Bangunharcana et al., 2021) and Coarse-to-Fine (CTF) strategy (Fig.2 right) (Tankovich et al., 2021; Yao et al., 2021; Shen et al., 2021; Wang et al., 2023). For U-shape Net, only volume at certain resolution is inputted and aggregated after downsampling, upsampling

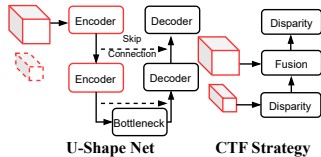

Figure 2: Two Structures.

and shortcut like UNet (Ronneberger et al., 2015). While CTF strategy models this as a refinement process that has the refined disparity map obtained from low resolution to higher resolution gradually in order, $\boldsymbol{D}^i = \mathcal{F}(\boldsymbol{V}^i|\boldsymbol{D}^{i+1})^2$. The former has no utilization of information from volumes at other resolutions, which is employed in later CTF but there is no deeper consideration of the volume-level scale invariance among volumes despite using scale invariance in feature extractor (feature level). In this paper, we first adopt the model strategy that gives up the disparity map generation during inference and utilizes the information across resolution at the same time.

**The General Inductive Bias for CNNs**  Generally recognizing, CNNs (LeCun et al., 1995) extract local features from the neighbor region, having the intrinsic IB in modeling locality. It is common to consider locality in conjunction with long-range dependencies, as they complement each other. Another critical topic in visual tasks is group equivariance(Goyal & Bengio, 2022) or scale invariance, which inspires the intra-layer fusion (Ronneberger et al., 2015; Lin et al., 2017) or inter-layer fusion(Szegedy et al., 2015; Zhao et al., 2017). Considering the scale invariance, (Wang et al., 2020) exchanges the information across resolutions in parallel, and (Zhang et al., 2022) fuses the information across resolutions only at the last resolution to meet the real-time requirements. We adopt the wisdom from the two methods and break down the dense information exchange into two processes.

**Long-Range Dependencies**  Attention is a wildly accepted method to capture long-range dependencies and can be divided into two categories: a normalization to reweight the target, rewritten as $\mathcal{N}(\mathcal{F}(X)) * X$, (Itti et al., 1998; Mnih et al., 2014; Hu et al., 2018; Woo et al., 2018) or non-local representation learning, usually known as self-attention, which computes the response at a token as a weighted sum of all tokens (Dosovitskiy et al.; Raghu et al., 2021; Wang et al., 2018; Chen et al., 2018). For the later attention mechanism, computational cost, no query-specific discovery and the deeper analysis of the relationship between convolution and self-attention (Cao et al., 2019; Ma et al., 2022) simplifies the operation as local-region operation with a global context, $\mathcal{N}(\mathcal{F}(X)) + X$. In MDLC, two forms ($\mathcal{N}(\mathcal{F}(X)) * X$ and $\mathcal{N}(\mathcal{F}(X)) + X$) are adopted by inherent volume structure without time-consuming Transformers introduced.

---

[2]For ease of representation, we use $\mathcal{F}(\cdot)$ and $\mathcal{N}(\cdot)$ to denote any function and normalization in this paper, which has no impact on the realization to the paper in fact.

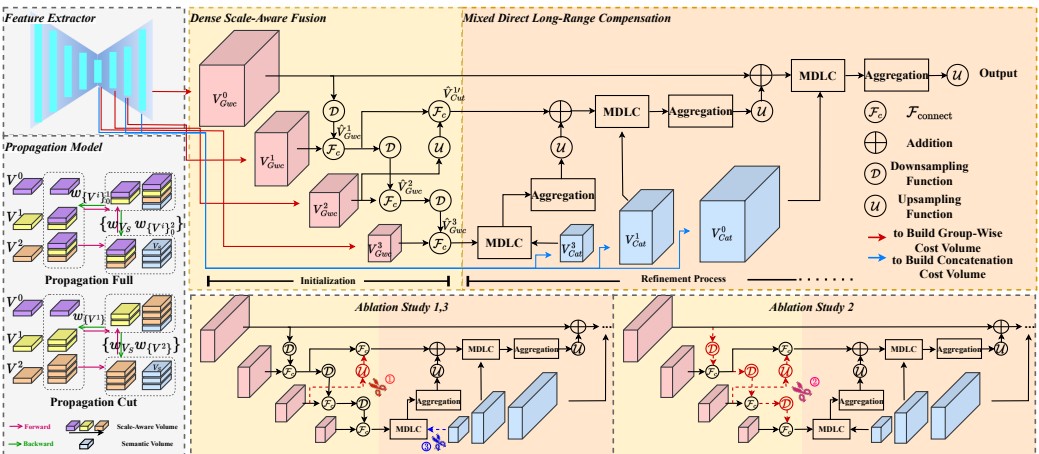

Figure 3: Our proposed network is constructed by Feature Extractor, Dense Scale-Aware Fusion and Mixed Direct Long-Range Compensation. Ablation Study 1,2,3 are the extra models we design for ablation study to validate effectiveness of general IB we introduce. Propagation Model simplify the signal propagation in our model and the ablation 3 model, respectively the Propagation Full/Cut.

## 3 METHOD

### 3.1 GENERAL INDUCTIVE BIAS AND USTEREO

Giving up disparity generation during inference and adopting the inter-layer fusion, UStereo is constructed based on DSF and MDLC, together with consideration to volume-level scale invariance and to capture long-range dependencies.

#### 3.1.1 DENSE SCALE-AWARE FUSION

To adopt volume-level scale invariance, we propose a simple recurrence fusion strategy DSF (Eq.2).

$$\hat{\boldsymbol{V}}^i = \mathcal{F}_{\text{connect}}(\boldsymbol{V}^i, \mathcal{D}(\hat{\boldsymbol{V}}^{(i-1)})) = \mathcal{F}_{\text{connect}}(\boldsymbol{V}^i, \mathcal{D}(\boldsymbol{V}^{(i-1)}, \boldsymbol{V}^{(i-2)}, \cdots)) \tag{2}$$

$$\hat{\boldsymbol{V}}_{cut}^i = \mathcal{F}_{\text{connect}}(\boldsymbol{V}^i, \mathcal{D}(\hat{\boldsymbol{V}}^{(i-1)}), \mathcal{U}(\boldsymbol{V}^{(i+1)})) \tag{3}$$

where $\mathcal{F}_{\text{connect}}$ denotes any connection function such as concatenation, addition, *etc.*, and $\mathcal{U}(\cdot)$ or $\mathcal{D}(\cdot)$ denote any up- or down-sampling function. $\hat{\boldsymbol{V}}$ and $\tilde{\boldsymbol{V}}$ denote $\boldsymbol{V}$ processed by Eq.2 or 5.

In this strategy, information at every resolution will be directly connected with information from higher resolution in initialization and subsequently at the refinement stage it will fuse with information from low resolution *i.e.* in Eq.2, we construct the relationship from bottom to up at $\boldsymbol{V}^i$ with all $\{\boldsymbol{V}^j\}_0^{i-1}$ and the information will aggregate from top to down at refinement process by inter-layer fusion. Due to DSF, information across resolution exchanges well, and the learning process will be enhanced when backpropagation (Wei et al., 2021) (in Fig.3 Propagation Full/Cut, DSF leads to information fusion and, according to chain rules, there will be a correlation between every volume). To lighten our model and maintain volume-level scale invariance, we skip the $2^{-3}$ resolution, and fuse information from $\boldsymbol{V}$ at $2^{-3}$ resolution to $2^{-2}$, that we replace $\hat{\boldsymbol{V}}^i$ with $\hat{\boldsymbol{V}}_{cut}^i$ by Eq.3.

#### 3.1.2 MIXED DIRECT LONG-RANGE COMPENSATION

Three key points prompt us to propose such a block: (i) intrinsic locality IB in CNNs needs a compensatory component to capture long-range dependencies but traditional Transformers are unfit for the real-time task; (ii) volume structure has the potential to capture context information and model long-range dependencies by constructing volume at lowest resolution and normalization; (iii) inter-layer fusion strategy can simplify the model structure alleviating the time cost in CTF strategy, and by the fusion, lower volume could become a global embedding after shallow stacks of convolutions.

Generally speaking, $\boldsymbol{V}_{Cat}$ focuses on semantic information and $\boldsymbol{V}_{Gwc}$ captures similarity. (Xu et al., 2022a) use a coarse map as filter weight generated in $\boldsymbol{V}_{Gwc}$ to restrain the $\boldsymbol{V}_{Cat}$. To simplify these

processes and inspired by multi-head in Transformers, we generate the multi-weight by the $\boldsymbol{V}_{Gwc}$ and use both kinds of information to generate a robust volume.

Besides the attention mechanism, another accepted approach to capture long-range dependencies and generate global context is by deep stacks of convolution(Wang et al., 2018). Due to the inherent volumes at low resolution *e.g.* $\boldsymbol{V} \in \mathbb{R}^{C \times D/32 \times H/32 \times W/32}$, global context can be easily obtained through shallow stacks of convolutions. Additionally inter-layer fusion strategy naturally encourages us to choose the global context as a compensatory component to capture long-range dependencies.

$$\boldsymbol{W}^i = \text{Softmax}_d(\text{Aggregation}(\hat{\boldsymbol{V}}_{Gwc}^i + \mathcal{U}(\tilde{\boldsymbol{V}}^{(i+1)}))) \tag{4}$$

Combine two above we propose a new compensation weight generation method (in Eq.4), constructed by two forms of compensatory components to capture long-range dependencies, *i.e.* $\mathcal{N}(\mathcal{F}(X)) * X$ and $\mathcal{N}(\mathcal{F}(X)) + X$. Then we propose a new block **MDLC** by dot product between the weight and the semantic information:

$$\tilde{\boldsymbol{V}}^i = \boldsymbol{W}^i \odot \boldsymbol{V}_{Cat}^i \tag{5}$$

### 3.1.3 MORE DETAILS ABOUT THE MODEL

**Disparity Regression and Output**   The confidence volume produced gives us matching confidence values of each disparity level for every pixel, which can be transformed into a probability distribution by taking a $\text{Softmax}$ across the disparity dimension. To generate disparity map $\boldsymbol{D}$, we both adopt the original disparity regression and top-K regression ($\mathcal{R}(\cdot)$) (Bangunharcana et al., 2021). We choose the up-sampling function $\mathcal{U}(\cdot)$ for the final output same as (Yang et al., 2020a).

$$\boldsymbol{D}_{x,y} = \mathcal{R}(\boldsymbol{V}_{x,y,:}, k) = \sum_{i=0}^{k} d_i \times \text{Softmax}(\boldsymbol{V}_{x,y,d_i}), \quad where \; \{\boldsymbol{V}_{x,y,d_i}\}_0^k = \textbf{Top}(\boldsymbol{V}_{x,y,:}, k) \tag{6}$$

The function $\textbf{Top}(\cdot, k)$ selects the top k values from the first input. Usually, $\boldsymbol{V}$ at final resolution is 3D without $\beta$ dimension.

**More Details**   We use MobileNet-V3 (Howard et al., 2019) as the encoder in Feature Extractor (in Fig.3), and the decoder is the simple block composed of single fractionally-strided convolution and Leaky ReLU (Xu et al., 2015). For simplification, $\mathcal{F}_{\text{connect}}(\cdot), \mathcal{U}(\cdot)$ and $\mathcal{D}(\cdot)$ are concatenation, linear interpolation and average pooling function respectively. We only use the simple Hourglass function for aggregation and design two extra simpler ones (see appendix A.2). For Feature Compress and Volume Compress in Fig.1, we use two layers of convolutions and a single respectively.

### 3.2 SPECIFIC INDUCTIVE BIAS AND DEEP SUPERVISION

In CTF strategy, there is disparity generated for fusion and supervision to the disparity will certainly improve the performance. To simplify the model structure, we adopt inter-layer fusion. Therefore, there is no disparity map generation during the inference of UStereo. Though volumes are constructed by Eq.1 related to specific IB and the model would finally create a confidence volume to predict the disparity map, representation in volume at lower resolution (volume during inference process) is vague, especially without disparity map generation. This raises our contemplation on specific IB, beyond the construction of the volume structure, whether specific IB or this structure has deeper underlying effects. To be specific, it remains a challenge to determine whether the volume at a lower resolution has the same impact as the confidence volume as well. Additionally, for the model with the same structure but different weight parameters and achieving better performance, the representation within volume is still an unresolved issue.

Inspired by the deep supervision (Lee et al., 2015), we design an in-depth study of specific IB and representation in volume at lower resolution. This study incorporates extra supervision to the volume at lower resolution, while maintaining the supervision at low resolution. By doing so, we aim to analyze the effects of this extra supervision on the overall performance. Therefore, the loss function in our work is constructed by a basic loss $\mathcal{L}_{Base}$ and extra loss $\mathcal{L}'$. $\boldsymbol{V}$ at resolution $= 2^{-2}$ and finial output are supervised in $\mathcal{L}_{Base}$, and $\mathcal{L}'$ is one of $\mathcal{L}_1, \mathcal{L}_2, \mathcal{L}_3$. $\boldsymbol{z}$ is the extra weight parameters in bypass deep supervision branch. $\text{Smooth}_{l_1}$ is smooth L1 in (Xu & Zhang, 2020).

$$\mathcal{L} = \mathcal{L}_{Base} + \mathcal{L}' \tag{7}$$

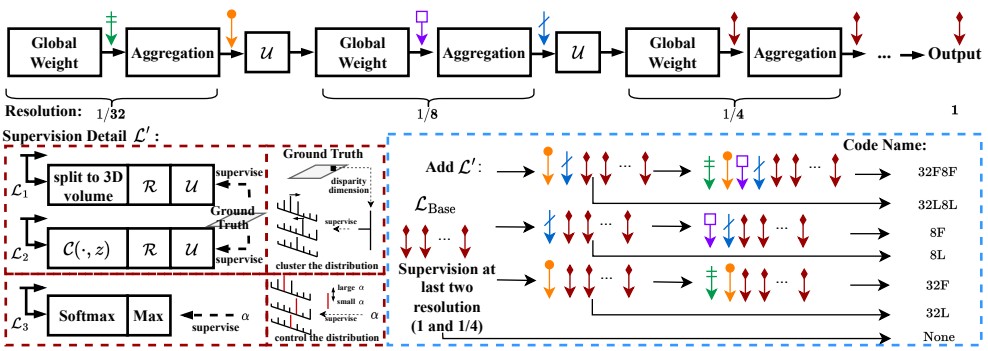

Figure 4: **Top**: the simplified UStereo with the necessary details. The location where we place colorful $\downarrow$ indicates where we add the supervision; brickred $\downarrow$ denotes $\mathcal{L}_{Base}$ and will be maintained during training. **Down**: Left side shows the detail $\mathcal{L}'$; Right shows the order in which we added extra deep supervision $\mathcal{L}'$ ($\downarrow\downarrow\downarrow$). We keep the same training times by repeating last supervision. XFXL is the code name, where 8/32 is about resolution and F represents supervision to weight and aggregation compared with L to aggregation

$$\mathcal{L}_{Base}(\boldsymbol{V}, \boldsymbol{D}^{gt}, \boldsymbol{z}) = \lambda \text{Smooth}_{l_1}(\mathcal{U}(\mathcal{R}(\boldsymbol{V}, k)), \boldsymbol{D}^{gt}) + \sum_{\boldsymbol{V}^i \text{ at res.}=2^{-2}} \lambda_i \mathcal{L}_2(\boldsymbol{V}^i, \boldsymbol{D}^{gt}, \boldsymbol{z}) \quad (8)$$

We suppose every 3D sub-volume along the $\beta$ dimension has the same impact. Therefore, We split the 4D volume into $N_{4/3}$ 3D volumes along the $\beta$ dimension and propose Eq.9 to to actively cluster the distribution within volumes. The set $\{\boldsymbol{V}^j\}$ denotes the volumes to be supervised in our training process (details are in Fig.4 and the number of elements in the set is related to different code name).

$$\mathcal{L}_1(\{\boldsymbol{V}^j\}, \boldsymbol{D}^{gt}) = \sum_j \lambda_j \text{Smooth}_{l_1}(\mathcal{U}(\mathcal{R}(\frac{1}{N_{4/3}} \sum_{n=0}^{N_{4/3}} (\boldsymbol{V}^j_{n,:,:,:}, k_j))), \boldsymbol{D}^{gt}) \quad (9)$$

A trainable deep supervision bypass branch is adopted as well, where sets of convolutions $\mathcal{C}(\cdot, \boldsymbol{z})$ with trainable parameters $\boldsymbol{z}$ are introduced but the structure of the model would not change at all. The learning process would make internal relationships vague unlike Eq.9. However, experiments show UStereo could obtain large improvement with deep supervision Eq.10 and it could become a strong tool to enhance the performance. To further explore the representation of the volume in enhancing model, we adopt 3 original statistic indicators to research characteristics of the disparity (representation).

$$\mathcal{L}_2(\{\boldsymbol{V}^j\}, \boldsymbol{D}^{gt}, \boldsymbol{z}') = \sum_j \lambda_j \text{Smooth}_{l_1}(\mathcal{U}(\mathcal{R}(\mathcal{C}(\boldsymbol{V}^j, \boldsymbol{z}'), k_j)), \boldsymbol{D}^{gt}) \quad (10)$$

$$\delta_{15} = \frac{\boldsymbol{\nu}_0 - \frac{\sum_{\text{last}50\%} \boldsymbol{\nu}_i}{N_{50\%}}}{\boldsymbol{\nu}_0 - \frac{\sum_{\text{last}10\%} \boldsymbol{\nu}_i}{N_{10\%}}} \quad \text{sum}_k = \sum_0^k \boldsymbol{\nu}_i \quad \mathcal{E} = \underset{k}{\arg\min}(\sum_0^k \boldsymbol{\nu}_i - \frac{1}{2}) \geq 0 \quad \boldsymbol{\nu} \in \text{Sort}_{\downarrow_d}(\boldsymbol{V}) \quad (11)$$

where $\boldsymbol{\nu}$ is a sorted vector from the disparity dimension of softmax-processed $\boldsymbol{V}$ in descending order, by $\text{Sort}_{\downarrow d}$, $\boldsymbol{\nu} \in \mathbb{R}^d$. $\text{last}X\%$ taking the last X percent of the vector, and $N_{X\%}$ corresponds to $X\%$ length of $\boldsymbol{\nu}$. $\delta_{15}$ is an indicator consider both the peak value and low values, and if it is large, it implies a relatively larger peak value and a smaller tail distribution. $\text{sum}_k$ and $\mathcal{E}$ serve a similar purpose.The former quantifies the amount required to reach 50%, while the latter provides a direct summary of the top k elements in the distribution. We set the k to 5 in this paper.

Eq.9 and 11 use $\boldsymbol{D}^{gt}$ in deep supervision based on the assumption from confidence volume, so probability distribution is clustered at the place of the matter in confidence volumes. To reveal the characteristics of the distribution along the disparity dimension within volumes, we propose an original deep self-supervision in Eq.12. The distribution of the disparity in the volume will be controlled by constant vector $\boldsymbol{\alpha}$ but has nothing to do with the distribution of confidence.

$$\mathcal{L}_3(\{\boldsymbol{V}^j\}, \boldsymbol{\alpha}) = \sum_j \lambda_j \|\text{Max}_\beta(\text{Softmax}_d(\boldsymbol{V}^j)) - \boldsymbol{\alpha}\|_{l_1}, \quad where \ \boldsymbol{\alpha} \in \mathbb{R}^\beta \quad (12)$$

| Method | SceneFlow | KITTI2012 | | | KITTI2015 | | | Time | Params | Dim |
|---|---|---|---|---|---|---|---|---|---|---|
| | EPE | 3 -noc | EPE all | EPE non | D1 -bg | D1 -fg | D1 -all | (ms) | M | 3/4 D |
| DeepPrunerFast | 0.97 | - | - | - | 2.32 | 3.91 | 2.59 | 40 | 7.46 | 4 |
| AANet | 0.87 | 1.91 | 0.5 | 0.6 | 1.99 | 5.39 | 2.55 | 62 | 3.93 | 3 |
| DecNet | 0.84 | - | - | - | 2.07 | 3.87 | 2.37 | 60 | 13.2 | 4 |
| BGNet | 1.17 | 1.77 | 0.6 | 0.6 | 2.07 | 4.74 | 2.51 | 54 | 2.97 | 4 |
| BGNet+ | - | 1.62 | 0.5 | 0.6 | 1.81 | 4.09 | 2.91 | 68 | 5.31 | 4 |
| CoEx | 0.69 | 1.55 | 0.5 | 0.5 | 1.79 | 3.82 | 2.13 | 33 | 2.7 | 4 |
| FastACV | 0.64 | 1.68 | 0.5 | 0.6 | 1.82 | 3.93 | 2.17 | 50 | 3.1 | 4 |
| Ours | **0.59** | 1.56 | **0.5** | **0.5** | **1.76** | 3.88 | **2.11** | 34 | 2.1 | 4 |

Table 1: Quantitative evaluation on the test sets of SceneFlow, KITTI 2012 and KITTI 2015 test sets. The runtime is tested on A6000. Volume-based method references: (Duggal et al., 2019; Xu & Zhang, 2020; Xu et al., 2021a; Yao et al., 2021; Bangunharcana et al., 2021; Xu et al., 2022b).

To ensure a fair comparison in §4.2 and following the recent trend of designing multi-stage training strategies, we adopt the training strategy in Fig.4. This strategy allows us to conduct related experiments under the same initial conditions, while also providing a beneficial initialization for the self-supervised approach during the training process.

# 4 EXPERIMENT

Our experiments are divided into three parts: first, compare with the state of the art (SOTA) volume-based method on four popular stereo datasets: SceneFlow Mayer et al. (2016), KITTI 2012 (Geiger et al., 2012), KITTI 2015 (Menze & Geiger, 2015) and SCARED (Allan et al., 2021), including virtual/real datasets and vast natural scenes/small abdominal cavity scene datasets; second, we design ablation studies to structure and IB introduced to our model; finally, following §3.2 we conduct study to the specific IB and the representation to volume.

The end-point error (EPE) and 1/2/3-pixel error (1/2/3pE) are reported on the dataset, where EPE is the mean disparity error in pixels and 1/2/3-pixel error is the average percentage of the pixel whose EPE is bigger than 1/2/3 pixel. The Mean Absolute depth Error (MAE) is reported on the SCARED dataset. The official metrics (*e.g.* D1-all) in the online leader board are reported as well.

For benchmarking, in spite of the simple structure, our model also achieves competitive performance in real-time stereo matching (Table 1), for example our model achieves significantly SOTA result in SceneFlow. Our model

| Method | SCARED MAE (mm) |
|---|---|
| J. C. Rosenthal | 3.75 |
| Trevor Zeffiro | 3.54 |
| Dimitris Psychogyios 1 | **2.33** |
| Dimitris Psychogyios 2 | 2.62 |
| Sebastian Schmid | 2.66 |
| MSDESIS | 2.85 |
| CFNet | 2.67 |
| RAFT-Stereo | 2.65 |
| Ours | 2.53 |

Table 2: The mean absolute depth error in mm for the average test dataset. References: (Allan et al., 2021; Psychogyios et al., 2022).

also has advantages on reference time and previous volume-based methods all have more parameters than ours. Comparison in Table 2 shows that our method exhibits robust generalization and achieves a good result on cavity scene datasets as well.

## 4.1 ABLATION FOR INTRODUCED GENERAL INDUCTIVE BIAS

Besides the basic ablation study, we also design extra models to discuss the IB. Details are below:

- Exp1: IB study for long-range dependencies, replacing $\mathrm{Softmax}$ with $\mathrm{Sigmoid}$ function in Eq.4.
- Exp2: IB study for scale invariance, removing entangled relationship with up-sampling in Eq.3.
- Exp3: IB study for scale invariance, removing dense fusion.

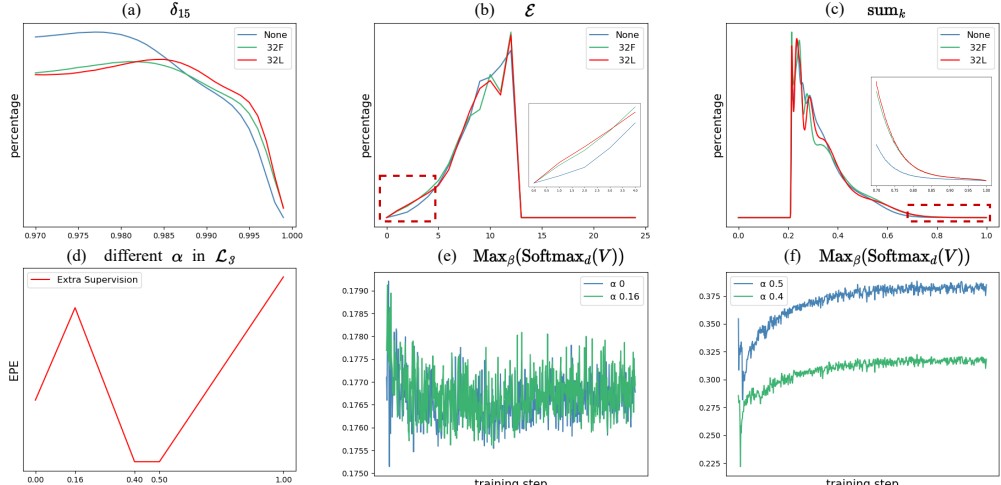

Figure 5: a-c show the distribution of statistic indicators we adopt, respectively $\delta_{15}$, $\mathcal{E}$ and $\mathrm{sum}_k$ in Eq.11. d shows the EPE result on Sceneflow when using different $\alpha$ in Eq.12 at 32L. e-f show how the mean of $\mathrm{Max}_\beta(\mathrm{Softmax}_d(\boldsymbol{V}))$ changes when training.

- Exp4: Ablation study to model structure, removing the entangled relationship between two volumes at the lowest resolution, to discuss the effect of information fusion. (Exp2 is related to Fig.3 Ablation 1, Exp3 is related to Fig.3 Ablation 2 and Exp4 is related to Fig.3 Ablation 3)
- Exp5: Ablation study to model structure, replacing two layers convolutions in Feature Compress to single, to discuss better representation learning for volume is good for model.

To explore the role of the long-range dependencies, we design the Exp1 replacing $\mathrm{Softmax}$ normalization with $\mathrm{Sigmoid}$ function. The result shows that without the long-range dependencies, the performance of the model gets worse. Exp4 can be viewed as an experiment to weaken the global context as well, the result of which also implies a well-processed global context has a great improvement to the performance. To discuss the IB in terms of the scale invariance, we design two extra models in Exp2 and Exp3. After removing the entangled relationship, both models get a worse performance. A simpler

| Experiment | EpE | 1pE | 2pE |
|---|---|---|---|
| Original | 0.593 | 6.82 | 3.72 |
| Exp1 | 0.618↓ | 6.92↓ | 3.81↓ |
| Exp2 | 0.609↓ | 6.82 | 3.73↓ |
| Exp3 | 0.611↓ | 6.96↓ | 3.77↓ |
| Exp4 | 0.614↓ | 7.05↓ | 3.83↓ |
| Exp5 | 0.598↓ | 6.76↑ | 3.70↑ |

Table 3: Ablation Studies.

structure in Feature Compress makes the model a weaker capacity for representation learning in volumes. Experimental results in Exp5 demonstrate a decline in the regression ability of the model, albeit with a decrease in the percentage of error points. This observation motivates us to further investigate the nuances of representational learning for the volume at lower resolution in future research endeavors, with the aim of enhancing the regression ability and overall robustness.

## 4.2 SPECIFIC INDUCTIVE BIASES AND REPRESENTATION FOR VOLUME

Under a strong assumption, we propose $\mathcal{L}_1$ (Eq.9). With the supervision from $\boldsymbol{D}^{gt}$, the disparity distribution in volume at lower resolution will cluster to the correct location. We add the extra deep supervision $\mathcal{L}_1$ to 32L and 8L respectively (see Fig.4). Experiments in Table 4 show that in spite of a strong assumption, the extra supervision could improve the performance of the model. This implies that the volume structure associated with specific IB functions as a confidence volume during inference.

To alleviate the problems with the above assumptions, we introduce trainable parameters in $\mathcal{L}_2$ (Eq.10). Experiments in Table 4 show extra supervision under different situations all have an improvement, which implies the power of the specific IB that volume structure serves as confidence volume during inference. Furthermore, experimental results demonstrate the importance of deep supervision at the lowest resolution, indicating the significance of a clustered distribution at the early stages, which further supports our hypothesis.

Additionally, we perform statistics on the volumes obtained at $2^{-3}$ resolution by model using extra deep supervision at 32L and 32F which have no direct relationship to $2^{-3}$ resolution. To be more specific, the statistical object is the disparity distribution of every single 'pixel' at $2^{-3}$ resolution volume, which is obtained through inference on the test dataset of the model. The sample size is around $10^9$. The three statistic indicators in Fig.5 all show that deep supervision facilitates concentration of the distribution, which implies a confidence volume and that concentrated distribution of the disparity would improve the performance. For $\delta_{15}$ and $\text{sum}_k$, distributions with extra supervision distribute at higher values. And for $\mathcal{E}$, distributions with extra supervision show that $50\%$ can be reached with less summation.

In Fig.5 d-f, we introduce Eq.12 to the supervision at 32L. Without $\boldsymbol{D}^{gt}$, the restricted volume will blindly learn a distribution trend under the control of $\alpha$. Fig.5 e and f show that under the restriction with Eq.12, larger $\alpha$ pushes a larger extreme value, and a small $\alpha$ makes a flattened distribution. Although the network is not limited to producing extreme values within the correct target, the results in Fig.5 d show that the relatively clustered distribution is better for the network to learn the pattern. This implies concentrated and appropriate distribution of the disparity helps the model learn the pattern. However, when adopting a strong constraint *i.e.* using extremely large $\alpha$ in Eq.12, the performance of the model will decline significantly.

| Strategy | $\mathcal{L}$ | k | EpE | 1pE | 2pE | 3pE |
|---|---|---|---|---|---|---|
| None | / | all | 0.593 | 6.82 | 3.72 | 2.72 |
| None | / | 3 | 0.578 | 6.47 | 3.60 | 2.65 |
| 32L | 1 | all | 0.590↑ | 6.81↑ | 3.71↑ | 2.69↑ |
| 8L | 1 | all | 0.591↑ | 6.85↓ | 3.73↓ | 2.71↑ |
| 32L | 2 | all | 0.581↑ | 6.75↑ | 3.66↑ | 2.66↑ |
| 32F | 2 | all | 0.582↑ | 6.77↑ | 3.69↑ | 2.68↑ |
| 8L | 2 | all | 0.591↑ | 6.81↑ | 3.71↑ | 2.70↑ |
| 8F | 2 | all | 0.588↑ | 6.75↑ | 3.69↑ | 2.69↑ |
| 32L8L | 2 | all | 0.585↑ | 6.77↑ | 3.69↑ | 2.68↑ |
| 32F8F | 2 | all | 0.591↑ | 6.82 | 3.72 | 2.72 |
| 32L | 2 | 3 | 0.573↑ | 6.38↑ | 3.54↑ | 2.59↑ |
| 8L | 2 | 3 | 0.573↑ | 6.40↑ | 3.55↑ | 2.61↑ |

Table 4: Specific Inductive Biases Studies for Volume by Deep-Supervision. Strategy relates to different supervision places and $\mathcal{L}$ is different from the loss function we use. 'all' means keeping k full in $\{d_j\}_0^k$ in Eq.6.

Experiments for Eq.9 as well as Eq.10 with statistic indicators related to representation within volumes demonstrate that a more concentrated distribution along the disparity within volumes enhances the model. Additionally, Experiments for Eq.12 imply the model could learn better with a concentrated and appropriate distribution within volumes. Therefore, the specific volume structure plays the role of confidence volume as well, and concentrated and appropriate distribution of the disparity within volume helps improve the learning capacity of the model. These could be the extensive role of the specific IB.

## 5 CONCLUSION AND LIMITATION

In this paper, we go deep into the general and specific IB of stereo matching. We introduce the general IB (scale invariance from CNNs and long-range dependencies, property from Transformers compensatory for intrinsic locality IB) into real-time stereo matching and hence develop a new model named UStereo based on DSF and MDLC. Additionally, we conduct an in-depth study of the specific IB and related representation. The volume structure is constructed related to specific IB and helps the model to finally get a confidence volume to predict the disparity map. Experimental results demonstrate that volumes during inference could be viewed as a confidence volume as well, and concentrated distribution of the disparity within volume helps the model learn the pattern better.

For the convenience of research, our model is a simple consideration to those IB. In addition, we have no further discussion to the model structure and there are several redundant structures. In future work, we will think about the design of the model and design a more effective and efficient model. Moreover, we will further generalize our ideas to other computer vision tasks.

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

## A  APPENDIX

### A.1  IMPLEMENTATION DETAILS

We implement our approach in PyTorch(Paszke et al., 2019) and use Adam(Kingma & Ba, 2014)($\beta_1$ = 0.9, $\beta_2$ = 0.999) as optimizer. We train our model on 1 NVIDIA A6000 GPU under our training strategy for 84 epochs. For SceneFlow, we use all training set (35454 stereo pairs) for training and evaluate on the standard test set (4370 stereo pairs). The learning rate starts at 0.001 and is decreased at 20th, 32th, 40th, 48th, 56th, 62th, 70th, 74th epoch at 1st stage and at 2th, 10th, 20th, 30th, 40th, 50th, 60th, 70th, 80th at the other stages. For fair comparison, we keep our all training processes four times. We set the coefficients $\{\lambda\}_0^3 = \{0.8, 1.0, 1.0, 1.0\}$ at first stage, and $\{\lambda\}_0^3 = \{0.8, 0.1, 1.0, 1.0\}$ at other stage. We set the extra loss coefficients in $\mathcal{L}'$ of Eq. 9 and 12 0.1, Eq.10 0.5.

For the KITTI dataset, we finetune the pre-trained SceneFlow model on the mixed KITTI2012/2015 training sets for 1000 epochs. The initial learning rate is 0.001 and decreased by half at 400th, 600th, 800th and 900th epoch. Then other epochs are trained on the separate KITTI 2012/2015 training set for benchmarking, with an initial learning rate of 0.0001.

The SCARED dataset comprises 7 training and 2 test videos featuring diverse porcine cadavers, captured using a da Vinci Xi surgical system, and ground truth are provided in form of pointclouds in the original left frame of reference. We pre-process the data the same as (Psychogyios et al.,

2022). During fine-tuning, we choose only to use the keyframes of datasets 1, 2, 3, 6, 7 and not the interpolated sequences. Training sets for 600 epochs and initial learning rate is 0.001 and decreased by half at 300th 400th, and 600th. We use the evaluation tool kit in (Psychogyios et al., 2022) as well.

## A.2 HOURGLASS STRUCTURE

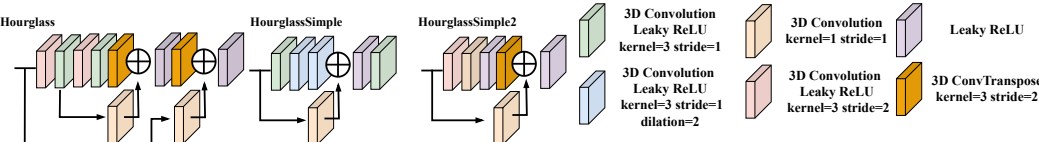

Figure 6: The architecture of the original Hourglass and two simple ones adopted in our study model.

In the inference order of our model, the Hourglass architectures used in aggregation are Hourglass-Simple, HourglassSimple, HourglassSimple2, HourglassSimple2, and the left are all Hourglass.

## A.3 SUPERVISION SET IN $\mathcal{L}_3$

For supervision at 32L, we use 0, 0.16, 0.4, 0.5, 1 for $\alpha$. The three numbers 0, 0.5, and 1 are sampled at equal intervals. The value 0.16 is generated from $1/d_{max}$, where $d_{max}$ is the length of $\nu$ and $1/d_{max}$ represents the lower number in the distribution. As the model achieves best performance with $\alpha$ equal to a relatively larger number 0.5, we also use 0.4 for $\alpha$.

## A.4 EXTRA EXPERIMENT FOR $\mathcal{L}_3$

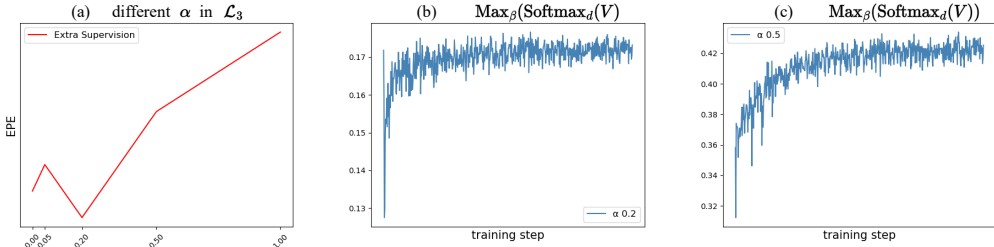

Figure 7: a shows the EPE result on Sceneflow when using different $\alpha$ in Eq.12 at 8L. b and c show how the mean of $\mathrm{Max}_\beta(\mathrm{Softmax}_d(\boldsymbol{V}^j))$ changes when training.

$d_{max}$ at resolution $2^{-3}$ is 4 times larger than $2^{-5}$, so $1/d_{max}$ at resolution $2^{-3}$ much smaller and even close to 0. In Fig.5 f, $sum_k$ is centrally distributed at 0.2, when k is set to 5. Therefore, for supervision at 32L we use 0, 0.05, 0.2, 0.5, 1 for $\alpha$. 0.05 is generated by $1/d_{max}$ as well. And we use 0.2 as the relatively larger $\alpha$. Supervision at 8L shares a similar conclusion as 32L and there 0.5 and 1 are large for the $d_{max}$, which is like '$\alpha$=1' when supervised at 32L.

## A.5 EXTRA EXPERIMENT FOR $\mathcal{L}_2$

We supplement the experiments with such case that k is set to half of the length of $\nu$ in Eq.6. In spite of a strong assumption, the extra supervision could improve the proportion of error pixels though experiment.

## A.6 TWO EXTRA STATISTIC INDICATORS AND MORE ANALYSIS TO REPRESENTATION

We also analyze the case for 8L. Obviously, when extra supervision is applied directly to 8L, the disparity distribution at 8L will be further clustered (see Fig.8). Fig.5 shows the distribution at $2^{-3}$

| Strategy | $\mathcal{L}$ | k | EpE | 1pE | 2pE | 3pE |
|---|---|---|---|---|---|---|
| None | / | 1/2 | 0.590 | 6.80 | 3.71 | 2.71 |
| 32L | 1 | 1/2 | 0.590 | 6.78↑ | 3.69↑ | 2.69↑ |
| 8L | 1 | 1/2 | 0.591↓ | 6.80 | 3.69↑ | 2.69↑ |
| 32L | 2 | 1/2 | 0.585↑ | 6.72↑ | 3.63↑ | 2.64↑ |
| 32F | 2 | 1/2 | 0.588↑ | 6.76↑ | 3.66↑ | 2.65↑ |
| 8L | 2 | 1/2 | 0.592↓ | 6.75↑ | 3.69↑ | 2.69↑ |
| 8F | 2 | 1/2 | 0.588↑ | 6.75↑ | 3.68↑ | 2.68↑ |
| 32L8L | 2 | 1/2 | 0.588↑ | 6.73↑ | 3.65↑ | 2.66↑ |
| 32F8F | 2 | 1/2 | 0.586↑ | 6.78↑ | 3.69↑ | 2.69↑ |

Table 5: Supplementary experiments for Table 4. 1/2 means keeping half in $\{d_j\}_0^k$ in Eq.6.

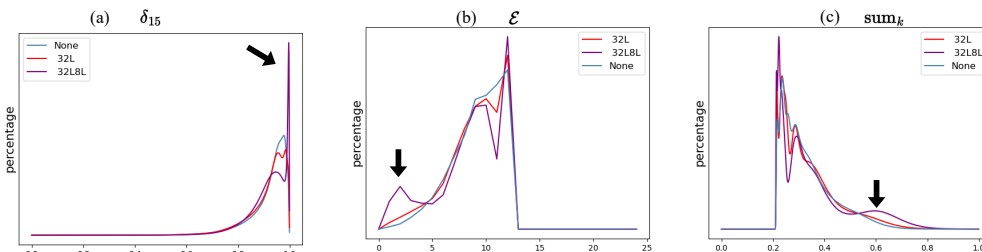

Figure 8: a-c show the distribution of statistic indicators for experiments in Table 4, respectively $\delta_{15}$, $\mathcal{E}$ and $\text{sum}_k$ in Eq.11.

when extra supervision is applied to 32L. Therefore, the characters of representation in Fig.5 are learned. Although in direct deep supervision to 8L and extremely clustered distribution, improvement is less than only to 32L and this is a common situation (Table 4 and Table 5).

For further analysis of the representation (distribution of disparity within volume), We introduce two novel parameters to examine the dynamics of the distribution. Specifically, we compute the absolute difference between the value points that encompass 50% of the distribution. We account for the presence of adjacent data points, and we ensure that the difference we ascertain are non-repetitive. Moreover, analysis is limited to the largest three absolute difference. This approach mitigates the influence of minor discrepancies at consecutive points on the overall results (in Fig.9). The detail procedure is in Procedure 1, and then we get two original statistic indicators $\delta_{50\%\text{top}}$ and $\delta_{50\%\text{mean}}$.

In Fig.10, the distribution of these two parameters has an obvious tendency towards 0. On the contrary, the distribution of adding deep supervision to the 32L/F clusters to the second peak. This phenomenon implies that despite direct supervision on 8L and greatly enhanced concentration, the distribution is relatively less differentiated. Singular deep supervision to 8L shares a similar tendency in Fig.11. This characteristic may also explain the poor performance of the model in Fig.5 d and Fig.7 when a relatively large supervision vector $\alpha$ is applied to the model, and therefore we conduct the experiments in §A.7.

## A.7   STRONGER SUPERVISION TO 8L

In order to verify the influence of the above distribution characteristics on the learning ability of the model, we further conducted the following experiments. We use a larger hyperparameter for loss coefficients at 8L from 0.5 to 0.8 in Eq.10, and hyperparameters for loss coefficients at 8L are 0.5 and 0.8 respectively. Experiments in Table 6 demonstrate that using a stronger supervision to 8L,

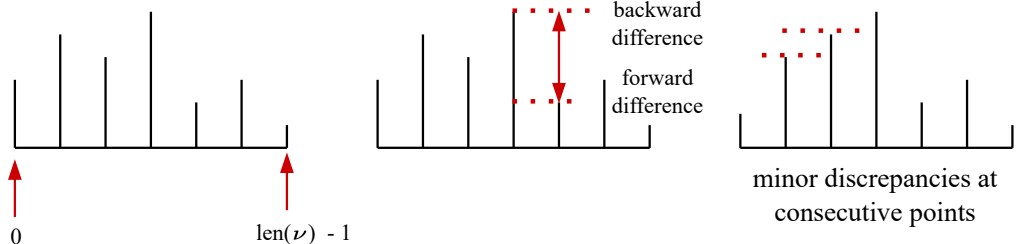

Figure 9: Visualization of the problems when using $\delta_{50\%\text{top}}$ and $\delta_{50\%\text{mean}}$.

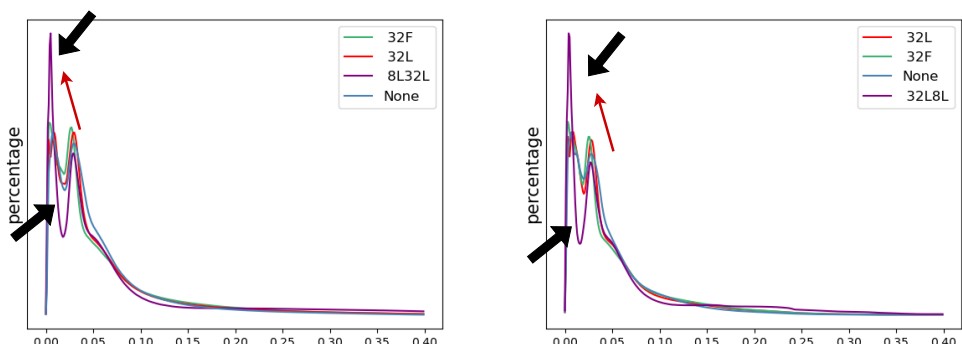

Figure 10: The pictures are respectively $\delta_{50\%\text{top}}$ and $\delta_{50\%\text{mean}}$. For the two peaks of the distribution, 32L8L is clearly clustered at the first crest. The second peak is also significantly down. And 32L/F clusters at the larger number.

the overall performance will be significantly worse, which is likely an influence by characteristics of the distribution in Fig.10.

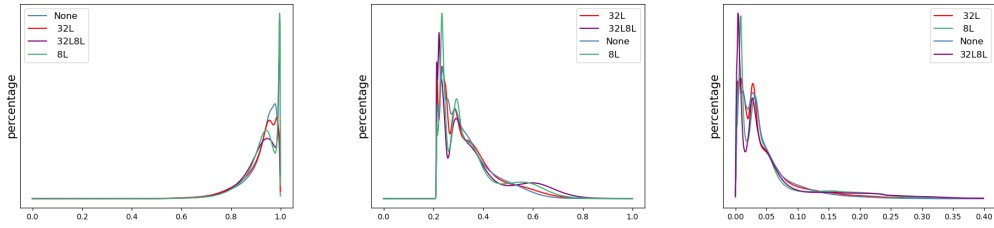

Figure 11: The pictures are respectively about distribution of $\delta_{15}$, $\text{sum}_k$ and $\delta_{50\%\text{mean}}$ for experiments in Table 4. 8L denotes when deep supervision is added to resolution $2^{-3}$. Compared with 32L8L, $\delta_{15}$, $\text{sum}_k$ seem to have a tendency to shift to the left for 8L, and 8L has a similar pattern as 32L8L.

---

**Procedure 1** Two extra statistic indicators $\delta_{50\%\text{top}}$ and $\delta_{50\%\text{mean}}$

---

**Input:** for any $\nu$ in $\text{Softmax}_d(V)$ (along disparity dimension)
**Output:** Two extra statistic indicators $\delta_{50\%\text{top}}$ and $\delta_{50\%\text{mean}}$
    // Stage 1. Generate $\nu_{50\%}$
1:  $\nu_\uparrow = \text{Sort}(\nu)$ // in increasing order
2:  **for** $i = 1$ to $\text{len}(\nu)$ **do**
3:    $\nu_{50\%} = \nu_\uparrow[-i :]$;
4:    **if** $\text{sum}(\nu_{50\%}) \geq 50\%$ **then**
5:      break;
6:    **end if**
7:  **end for**
8:  Find the original index in $\nu$ for $\nu_{50\%}$, then we get $\mathbf{ind}_{\nu_{50\%}}$;
9:  $\mathbf{ind}_{\nu_{50\%\uparrow}} = \text{Sort}(\mathbf{ind}_{\nu_{50\%}})$ // in increasing order
10: Initialize $\boldsymbol{\delta}$ list = [];
11: Set $\theta_{\text{adjoin}} = 1$;
    // Stage 2. Get Target Parameters
12: **for** $j = 1$ to $\text{len}(\nu)$ **do**
13:    **if** $\mathbf{ind}_{\nu_{50\%\uparrow}}[j] > 0$ **then**
14:      $\text{abs}(\nu[\mathbf{ind}_{\nu_{50\%\uparrow}}[j-1]] - \nu[\mathbf{ind}_{\nu_{50\%\uparrow}}[j]])$ append to $\boldsymbol{\delta}$;
15:    **end if**
16:    **if** $\mathbf{ind}_{\nu_{50\%\uparrow}}[j] < \text{len}(\nu) - 1$ and $\mathbf{ind}_{\nu_{50\%\uparrow}}[j+1] - \mathbf{ind}_{\nu_{50\%\uparrow}}[j] > \theta_{\text{adjoin}}$ **then**
17:      $\text{abs}(\nu[\mathbf{ind}_{\nu_{50\%\uparrow}}[j+1]] - \nu[\mathbf{ind}_{\nu_{50\%\uparrow}}[j]])$ append to $\boldsymbol{\delta}$;
18:    **end if**
19: **end for**
20: $\boldsymbol{\delta}_\uparrow = \text{Sort}(\boldsymbol{\delta})$ // in increasing order;
21: **if** $\text{len}(\boldsymbol{\delta}_\uparrow \geq 3)$ **then**
22:    $\boldsymbol{\delta}_\uparrow = \boldsymbol{\delta}_\uparrow[-3 :]$;
23: **end if**
24: $\delta_{50\%\text{top}} = \boldsymbol{\delta}_\uparrow[-1]$ ;
25: $\delta_{50\%\text{mean}} = $ average of $\boldsymbol{\delta}_\uparrow$ ;

---

| Strategy | $\mathcal{L}$ | k | EpE | 1pE | 2pE | 3pE |
|---|---|---|---|---|---|---|
| 8L | 2 | all | 0.596 ↓ | 6.81 | 3.72 | 2.71 |
| 8F | 2 | all | 0.589↓ | 6.77 | 3.71 | 2.70 |
| 328L | 2 | all | 0.592 ↓ | 6.77 | 3.70 | 2.69 |
| 32F8F | 2 | all | 0.591 ↓ | 6.78 | 3.69 | 2.68 |
| 8L | 2 | 1/2 | 0.594 ↓ | 6.75 | 3.69 | 2.69 |
| 8F | 2 | 1/2 | 0.590 ↓ | 6.76 | 3.67 | 2.67 |

Table 6: Using larger hyperparameter for the loss coefficient at 8L changing from 0.5 to 0.8 in Eq.10.

