# OpenReview forum: "Going Deeper with General and Specific Inductive Bias for Real-Time Stereo Matching"
_ICLR.cc/2024/Conference — Submitted to ICLR 2024_

### Official Review · Reviewer_ocBs · 2023-10-28

**Soundness:** 3 good
**Presentation:** 1 poor
**Contribution:** 2 fair
**Rating:** 3
**Confidence:** 4

**Summary:**

In this paper, the authors focus on designing stereo matching networks with both fast speed and robust generalization ability. They achieve this goal by thinking of the Inductive Bias (IB) in CNNs and Transformers, as well as the specific  IB in cost volumes. Specifically, they consider the scale invariance and long-range dependencies for real-time stereo matching and propose a new model UStereo. Through the study of specific IB of volume, they show the volume structure can be viewed as a confidence volume, which helps to learn better.

**Strengths:**

1. This paper builds a new stereo matching network UStereo that can achieve real-time inference and great accuracy.
2. They propose to think about the specific IB in cost volume representation, which is interesting.

**Weaknesses:**

1. The writing needs improving. Introducing the IB concept makes it hard to understand what the authors claim. I wonder why it needs to introduce IB instead of simply exploring scale invariance and long-range dependencies.
2. The results of the KITTI benchmarks are not impressive. Among real-time methods, we find CGI-Stereo, HITNet, and Fast-ACVNet+ get better performances.
3. There is a lack of evaluating the generalization ability in common settings. This paper claims UStereo has robust generalization, however, the common setting for generalization evaluation is on the KITTI, Middlebury, and ETH3D datasets.

**Questions:**

1.  I still wonder what is the specific inductive bias in the volume representation. I think using the cost volume as confidence representation has been explored, especially in the are of Muli-view Stereo. So I want to know if there is any other specific IB proposed in this paper.

---

### Official Review · Reviewer_DNk5 · 2023-10-31

**Soundness:** 2 fair
**Presentation:** 1 poor
**Contribution:** 2 fair
**Rating:** 3
**Confidence:** 4

**Summary:**

This paper proposes a novel stereo matching network (UStereo) with two main modules (Dense Scale Aware Fusion and Mixed Direct Long Range Compensation) to capture the generic bias information. The Dense Scale Aware Fusion (DSF) solves the volume-level scale invariance IB, and the Mixed Direct Long Range Compensation (MDLR) solves the long-range dependencies IB. In addition, a specific IB and its representation is studied in the paper and the paper proposes three novel statistic indicators to concentrate and control the distribution of the disparity within the volume. Experimental results show that the proposed method achieves fast speed and show robust generalization capabilities across datasets.

**Strengths:**

+ The proposed UStereo network utilizes general and specific IB to consider the volume-level scale invariance and long-range dependencies.

+ The proposed method has small number of parameters and fast computation. This is due to a carefully designed hierarchical stereo matching network, such as skipping the resolution in the middle.

+ Experimental results have been done in various challenging dataset including the SCARED which is the small abdominal cavity scene datasets.

**Weaknesses:**

- The claim of the proposed method is robust in generalization is not well justified. It finetunes the pre-trained model in the target dataset and is evaluated in the target dataset. In [A], they claim to be robust in generalization as it is not finetuned in the target dataset.

- The ablation study is not well-written. It is difficult to follow the decision process of each module (DSF & MLDC). It is better to focus on each module separately. It is recommended to follow Table 2 in [A] to show better ablation study. It is unclear why the softmax normalization is replaced by Sigmoid function to validate the long range dependencies. Wouldn't it better to remove the MDLC module for the ablation study as in Exp3?

- The goal of the specific IB is to perform the deep supervision in the lower resolution so that the distribution of the lower-level cost volume can be supervised by the distribution of ground truth disparity. The idea of deep supervision is not new because it has been done in [B] as well. However, it is unclear how the proposed additional supervised loss can lead to better performance.

- The performance improvement seems to be incremental, while the justification of the proposed method is lacking due to unclear experimental results. More exhaustive ablation study is required to justify the performance, especially comparison with various layer fusion methods, global and local attention stereo matching methods.

- The motivation of the problem is unclear. There is no experimental based justification that the current stereo matching method performance is insufficient due to lack inductive bias in the network design. Selecting the state-of-the-art methods and showing the problem quantitatively might be better approach to show the motivation.

Additional references
- [A] Domain Generalized Stereo Matching via Hierarchical Visual Transformation, CVPR 2023
- [B] CFNet: Cascade and Fused Cost Volume for Robust Stereo Matching, CVPR 2021

**Questions:**

- How is the performance of the proposed method when it is not fine tuned in the target evaluation dataset?
- Wouldn't it better to remove the MDLC module for the ablation study as in Exp3?
- What is the baseline network and its performance?
- As the proposed method focuses on the DSF and MLDC, how are the performance of those methods when they are integrated with current state-of-the art methods?
- Is the inductive bias the reason that the performance of SOTA stereo matching methods insufficient? How can we justify the question?

---

### Official Review · Reviewer_J9Ts · 2023-11-02

**Soundness:** 2 fair
**Presentation:** 2 fair
**Contribution:** 2 fair
**Rating:** 3
**Confidence:** 3

**Summary:**

A new ``UStereo'' approach for real time stereo matching.

**Strengths:**

A new method  UStereo based on DSF and MDLC.

**Weaknesses:**

Very poorly written paper, should be rejected for now based on writing quality alone. Could have a contribution but would need to be rewritten. Suggestions.

1. Review the literature on deep learning for stereo matching
2. Explain how your chosen inductive bias has not been seen in the literature
3. Note the use of attention mechanisms to model long range dependencies
4. Introduce your model of stereo. Explain the role of scale etc.
5. Showcase your results.
6. Drop the real-time aspect.

**Questions:**

The paper is very weirdly written. Why is it framed as a new inductive bias and why the emphasis on real-time?

I loved the sentence: ``Attention is a wildly accepted method to capture long-range dependencies''. The authors probably meant to say ``widely'' but wildly also fits.

**Details Of Ethics Concerns:**

N/A. Note that as a classical liberal, I object to the term ``Potentially harmful insights, methodologies and applications.'' Ideas cannot cause harm. This is bad wokeness (as opposed to good wokeness) interfering with research.

---

### Official Review · Reviewer_CP2t · 2023-11-09

**Soundness:** 2 fair
**Presentation:** 1 poor
**Contribution:** 2 fair
**Rating:** 1
**Confidence:** 4

**Summary:**

This paper proposes a  new  architecture  for stereo,  based on  the  concept  of acheiving a  more  general  inductive  bias (IB).
Considering the omnipresent  use  of a  correspondance volume as  an example  of  specific IB,  the paper intends  to
provide  a  better architecture that feature scale-invariance and long-range  dependencies,  as well as fast inference.

This architecture proposes two  new modules  (DSF and MDLC),  and three "statistic indicators".

**Strengths:**

The concept of Dense Scale-aware fusion is interesting,  although it is hard to  understand  how  it  relates  to  more
traditional approaches to scale management in stereo.
The mention (p.4) by the  authors
> Due to DSF, information across resolution exchanges well, and the learning process will be enhanced when backpropagation.
suggest that  experiments  should  be  devised to explicitely demonstrate this property.

The concept of  compensating long-range dependecies without transformers is  interesting,  but  would  deserve  more  in-depth
study,  to  illustrate exactly how it related/outperform  transformers as well  as current  multi-resolution CNN.

The results look adequate, on par with current fast stereo methods.

**Weaknesses:**

The paper is very hard to read, both because of  the writing  quality  and  the  lack  of  clarity.
Some sentences make very little sense,  possibly because this. As  an exemple (p.6):
> The learning process would make internal relationships vague unlike Eq.9

Figure 1, 2 and 3 are not clear enough  to help  understand the  architecture.


The ablation study does  not  clearly illustrate  how the proposed  modules actualy get better performance. Also, many decisions
in the architecture are not  justified and  would  deserve  to  be  part  of  the  ablation  study (ex: using Top-5 for  disparity regression)


Eq.12 suggests that the Max of a Softmax, which is a probability,  will be  minimized. Can  this  be justified?


**Results** : Table 1 provide performance on  KITTI12 and  KITTI15, but **the  paper  results  are  not available in  the  online  dashboard**.  This is very  important  and must be fixed if  the  results  are  to  be  taken  seriously.

Also,  Table 1 result for BGNet+ KITTI2015 D1  result  is not  2.91, but  2.19.
Moreover, many results of  Table are  cited  from  published  papers,  but the actual  online results are  better  (see  CoEx and FastACV). Even if this is not a major  problem, this disrepency  should  be  mentioned by the authors.

Table 4 is  hard  to  understand, as well as the  coding  (32L,8L,32F, etc.)  used  for deep extra supervision.

The two extra statistics  indicators ($\delta$ 50%top and $\delta$ 50%mean) are  not  well justified.
Maybe an ablation study would  help  justify them.
The pseudocode (p.17) is not useful, as  these  indicators are  very easy to  compute.

Many Spelling and formulations problems (here  are a few, but there are many more):
 - p.1 invarience
 - p.1 for almost vision tasks
 - p.2 knowledgement
 - p.3 wildly ->  widely
 - p.5 our contemplation
 - ...

**Questions:**

Beside what has been mentionned above:

It would be important to  clarify what is meant by  "scale-invariance", given that in  stereo  disparity will  vary with  scale. Maybe "scale-equivariant" would  be more  appropriate?

The role of *deep supervision* (p.5) is not clearly defined,  but Fig.4 suggests a very large number of losses, and possibly
a  very important role. Can  this be  clarified and  be  part  of a  specific ablation  study?

---

### Official Review · Reviewer_1LXD · 2023-11-10

**Soundness:** 3 good
**Presentation:** 3 good
**Contribution:** 3 good
**Rating:** 3
**Confidence:** 5

**Summary:**

The paper developed a model named UStereo to introduce the general IB to stereo matching. It adopt inter-layer fusion to break down volume-level scale invariance to a recurrence strategy in initialization for information at low resolution and refinement process for the high, which further extends to capture long-range dependencies after shallow stacks of convolutions and normalization without time-consuming Transformers. Additionally, to reveal the role that the volume structure constructed by specific IB plays during inference, it proposed the first-time in-depth study of volume at low resolution through varying degrees of restraint as well as 3 original statistic indicators to reflect the characteristics of representation within volumes.

**Strengths:**

+ A new model called UStereo, where disparity generation during inference is given up and replaced by direct inter-layer fusion.

+ An in-depth study of the volume structure at low resolution, deep into the role of specific IB.

+ UStereo achieves competitive results to other volume-based algorithms with fast speed and shows generalization capabilities across virtual and real datasets, as well as diverse natural scenes and small abdominal cavity scenes.

**Weaknesses:**

There are major issues with the paper writing. While the proposed ideas may sound, it used a rather indirect way to make the presentation.

The evaluations are insufficient. It did not perform sufficient evaluation in demonstrating the superority of the proposed solution in terms of accuracy, running time and memory.

**Questions:**

Please refer to my questions as listed in the Weaknesses section.

---

### Meta-Review · Area_Chair_oyAY · 2023-12-06

**Metareview:**

The authors claim to develop a novel model named as UStereo to introduce the general inductive bias to stereo matching. Based on the writing, limited receptive field is one of inductive bias by CNN models and the authors propose inter-layer fusion to break down volume-level scale invariance. Although this idea might be new, the overall presentation of the paper and the experimental validation are weak. This has been pointed out by the reviewers consistently. Therefore, I cannot consider this paper.

**Justification For Why Not Higher Score:**

The paper is not well written and lacks of clarity in multiple aspects.

**Justification For Why Not Lower Score:**

NA

---

### Decision · Program_Chairs · 2024-01-16

Reject